# Gamit! Icing on the Cake for Mathematics Gamification

Elvira G. Rincon-Flores [1],[*], Brenda N. Santos-Guevara [2], Lizette Martinez-Cardiel [3], Nadia K. Rodriguez-Rodriguez [4], Hernan A. Quintana-Cruz [4] and Alberto Matsuura-Sonoda [5]

1 Institute for the Future of Education, Tecnologico de Monterrey, Monterrey 64849, NL, Mexico
2 Center for Professional Development and Educational Partnership, Campus Monterrey, Tecnologico de Monterrey, Monterrey 64849, NL, Mexico
3 Department of Applied Pedagogy, Universidad Autónoma de Barcelona, Bellaterra, 08193 Barcelona, Spain
4 Carrera de Ingeniería de Sistemas, Universidad de Lima, Santiago de Surco 15023, Peru
5 Innovación Educativa, Universidad de Lima, Santiago de Surco 15023, Peru
* Correspondence: elvira.rincon@tec.mx

**Abstract:** Gamification has permeated education as a strategy to improve the teaching-learning process. Research shows that gamified reward systems based on badges, leaderboards, and avatars modifies the learning environment and student attitudes. This research aimed primarily to assess the change in attitude towards mathematics in high school students through a gamified methodology involving a reward system managed through a web platform called Gamit! This platform was developed by professors from two Latin American universities to manage gamification in a way that ensured that the anonymity of the class rankings was maintained. A mixed (QUAN-Qual) and quasi-experimental methodological approach was used for this study; two questionnaires were applied to 454 high school students and a focus group was performed with a group of seven professors. The quantitative analysis was processed with SPSS and consisted of ANOVAS and post hoc tests for more than two samples, while the focus group analysis was performed through inductive analysis. Results show benefits for professors and learners. Students improved their attitudes toward mathematics, reducing anxiety and improving willingness, while professors found a dynamic and optimal way to manage gamification on Gamit!

**Keywords:** gamification; innovative education; digital leaderboards; attitude toward mathematics; math education

## 1. Introduction

Over the years, there has been a need to promote lifelong learning as it recognizes the importance of establishing teaching experiences that have a positive effect on people's lives. If it is possible to foster a taste for learning in addition to achieving the development of useful and lasting skills and knowledge, we will have achieved educational success.

One of the concerns in the teaching of mathematics is the low rate of academic success [1]. Although some of the causes are attributed to the process of teaching, this lack of success is also due to the lack of didactic strategies that promote active learning [2,3]. In this sense and because of recurrent practices and research, gamification based on a reward mechanism is one of the educational tools that favors active learning [4–7], and in the literature we can find a vast number of studies that support this statement [8–11]. In 2017, a methodology based on the mechanics of rewards was applied in a Mexican university, which consisted of rewarding the knowledge, attitudes, values, and skills of the students through diverse types of badges. These were initially placed on a physical board, and the students were represented by an avatar whose image was received by the professor via email. The professor was then given the task of updating and pasting the badges awarded to the students to the physical board on a weekly basis; although the results of gamification were rewarding, the physical handling of the board represented extra work for the professor.

Collaboration between universities is essential to generate innovative methods in academic environments [12]. In that sense, the Tec de Monterrey of Mexico and the Universidad de Lima from Peru have worked to develop synergies since 2019 to implement Gamit! Gamit! is an interactive gamification digital platform designed for higher education professors and students. It is based on a reward system that recognizes student performance and skills using badges, avatars, and leaderboards. The team of Mexican researchers proposed the methodology for Gamit!, while the Peruvian researchers proposed the implementation of Gamit! on a digital web platform that would facilitate the management of gamification for professors and students.

The usage of technological tools usage in education has been rising, especially during the pandemic. These tools usually help professors and students be more connected and have common resources to find information. That is why we planned to build a web platform that helps instructors adopt gamification techniques and lets them use it during their classes more easily without creating extra work. The key features that the platform implements are divided into two parts: the ones that are tailored to instructors and the ones that focus on students' needs. In the case of instructors, they can manage their leaderboards, assign badges to students, and visualize a performance report. Students can register in a class, select an avatar, and see their performance within a class on the leaderboard.

At the communication level, it was important to find a balance between the institutional image of the two universities. For this, several graphic design tests were carried out, and the use of a gradient color that combines both institutional colors was selected: the blue of the Tec de Monterrey and the orange of the Universidad de Lima. This resource was used as a visual background for the platform and allows Gamit! to give its own character.

Before presenting the methodology and results, a review of the literature is shown in which the topics of attitudes toward mathematics and educational gamification are addressed with the purpose of offering a conceptual frame.

### 1.1. Why Are Attitudes toward Mathematics Important in Learning?

Mathematics is a subject that is still present in the current era, and surely every professor of this subject fervently believes that the basic learning of mathematics should be a natural process since it is inherent to any area that a high school student should know regardless of the career, he/she is about to choose. However, high school students' attitudes toward math remains a challenge for many professors, at least in Latin America. In Mexico, 66% of students who complete a high school education have insufficient curricular mathematical knowledge [13], and on average, Latin American students were at the lowest level in the PISA test (Programme for International Student Assessment) according to results published in 2019 [14].

The attitude towards mathematics is defined as the affective response, whether positive or negative, which implies a personal commitment and generates a behavior. In other words, it is how the student responds to learning mathematics [15,16]. Attitudes are composed of three elements: cognitive, affective, and behavioral. *Cognitive* consists of ideas, perceptions, beliefs, and opinions; *affective* consists primarily of feelings, such as those of liking or disliking; and *behavioral* consists of a visible reaction, tendencies, dispositions, or intentions [17–19].

According to Auzmendi [20], attitudes towards mathematics can be characterized in five dimensions: *anxiety*, the feeling of fear that the student manifests before the subject; *enjoyment*, the joy that the mathematical work provokes in the subject; *utility*, the usefulness that the student perceives that the subject may have in his future professional life; *motivation*, towards its study and use; and *confidence*, the feeling of security in math ability. The study [21] found that the meanings that students share about attitudes towards mathematics are participatory, democratic, and inclusive processes that allow them to express anxiety, confidence, utility, motivation, and an interest in mathematics.

Math anxiety is an important problem and is insufficiently addressed in the teaching of mathematics, at least in Latin American countries, because students who suffer from it

to a lesser or greater degree receive surge of intrusive thoughts when they try to solve a mathematical problem; there is even a negative correlation between mathematical anxiety and math performance [22]. Similarly, in another study, they confirmed the hypothesis proposed by others that attitudes towards mathematics are related to performance, particularly anxiety and enjoyment being significant predictors of math performance [23].

The topic of attitudes towards mathematics becomes more valuable in adolescents, since during high school they take many mathematics courses, and, in addition, their physiological stage is complex [24]. For these reasons, in the present study, it was decided to measure the impact of gamification based on the mechanics of rewards in attitudes toward mathematics with the purpose of offering the scientific and academic community the possibility of improving the math teaching-learning process.

### 1.2. Gamification to Ice the Cake for Learning

Gamification emerged as one of these pedagogical strategies giving rise to the use of game elements in educational scenarios that in principle are not for play. Research on teaching techniques and learning motivation shows that a pleasant, comfortable, safe, and enjoyable environment [11] is a good start to promote lifelong learning and achieve encouraging results for educational systems. Even though students may perceive learning as boring [25], gamification is able to enhance the learning process. The authors view gamification as offering experience creation and recreation, that can produce behavior changes with a sense of autonomy and dominance [26].

García et al. [27] points out that learning has typically been focused on passing exams more than long-life learning, incentivizing extrinsic motivation instead of intrinsic motivation. Gamification looks for active learning, increasing intrinsic motivation [28].

The starting point is to ask ourselves what the objective of gamifying is, either in whole or in part, and why gamification could help us to achieve the desired objective. Once this becomes clear, we will have to make a detailed and intentional design that frames the gamification that must necessarily be related to the course.

The danger of choosing a disconnected or unfamiliar frame for gamification can last not only as an overexertion without appropriate results but can even cause results opposite to those desired [29,30]. It is possible that gamification generates additional stress or work as a distractor rather than being a benefit when there is no relationship or support for the subject used.

Thus, if we have already decided that gamification is the technique that can support us with our learning objectives, it is necessary to think carefully about what our narrative will be. A narrative is one of the most principal elements concerning these models [31], because it sets the context and determines the connection between the gaming experience and users' interest and engagement. The narrative can be seen as the theme in which each gamification is based.

Badges and leaderboards are the most frequently used gamified elements, serving as enablers for competition, motivation, and feedback providers [32]. Badges are a representation of achievement and are earned by completing tasks. They also represent interests or affiliation [33]. Leaderboards work on different tracks; they show scores, rankings, avatars, progress, and, more than anything else, they contribute to engage members through the achievement's socialization, in this case, for learning purposes and collaboration [30].

Interactions are another principal element for learning and are increased by gamification [29]; they give a sense of belonging and partnership needed to maintain behaviors and motivation [32].

On the other hand, design is a key element in gamification, and not only should the narrative be interesting and meaningful for students, the experience and look are what makes gamification attractive [34] and lasting.

## 2. Materials and Methods

### 2.1. Methodology

The methodological approach was mixed and quasi-experimental, of the QUAN-Qual type (with a dominant quantitative element), with the aim of complementing the quantitative results with the qualitative ones [35]. The sample consisted of 454 students, which was representative (N = 2400, $\propto$ = 0.05, *n* = 332). Regarding professors, seven participated, and four of them have been using the gamification methodology since August 2021. The present study was carried out during the August-December 2022 semester. The research objective was to assess the change in attitude towards mathematics in high school students through a gamified methodology involving a reward system managed through a web platform called Gamit!

The study was conducted through the following research questions:

1. How did the reward-based gamification methodology influence high school students' attitudes toward math?
2. What is the relationship between the gamification methodology based on a reward mechanic and the attention, engagement, and resilience dimensions on the part of students and professors?
3. What is the perception of the usability of the Gamit! platform by students and professors?

The purpose of the first question was to answer the main objective of the research, and the second provided information on the aspects of attention, motivation, resilience, and engagement from the perspective of both actors, students, and teachers; these are key elements in the teaching-learning process and at the same time are related to gamification. Finally, the third yielded information about the user experience, which will improve the platform.

To answer these questions, a questionnaire was given to students before and at the end of the courses on attitudes towards mathematics [20], as well as a questionnaire on the dimension's usability, engagement, and resilience, whose Cronbach's alphas were 0.822, 0.903, 0.915 and 0.929, respectively. Both instruments correspond to a Likert-type scale but are continuous from 1 to 10 points. The analyses performed were processed with SPSS and consisted of ANOVAS and post hoc tests for more than two samples.

In addition, a focus group was created for the professors in which the same dimensions as with the students were addressed, and we applied an inductive analysis. The focus group description was the first presented topic by the moderator, as well as mention of the dimensions and students' questionnaires. The conversation began by getting to professors' experiences with gamification and the way in which each one implemented it. Once all had the main idea and the referenced experience as a starting point, the dimensions were approached in the same order that they appeared on the students' questionaries. The analysis was performed by another person and validated by the moderator and the observer.

### 2.2. Results

First, we will present the sociodemographic results. Of the 454 students who participated in the study, 49.8% were male, 48.9% were female, and 1.3% of students preferred not to say. Fifty-five percent (55%) of young people had a scholarship, while 83.5% attended the bicultural program (students who take courses in Spanish and English) and the rest attended the multicultural one (students who study subjects in more than two languages). On the other hand, 55% of the students attended the first semester, 29% attended the third semester, and 15.91% attended the fifth semester. The ages of pupils ranged from 14 to 18. Figure 1 shows the distribution.

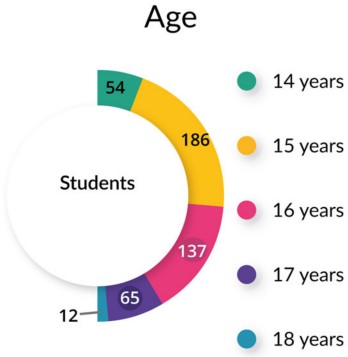

**Figure 1.** Students' age.

As we can see, most of the students were between the ages of 15 and 16 years.

### 2.2.1. First Question Results

We will now address the first question: How did the gamification methodology, based on a reward mechanism, influence high school students' attitudes toward mathematics?

The Auzmendi Attitudes Towards Mathematics test [20] was applied, which is formed by the dimensions of anxiety, pleasure, utility, motivation, and confidence. Additionally, the procrastination dimension of another instrument [36] was adapted and added. Table 1 shows the overall results.

**Table 1.** Overall Attitude toward mathematics results.

| Dimension | Pre-Test | Post-Test | Significance $\propto$ |
|:---:|:---:|:---:|:---:|
| Anxiety | 6.01 (2.2) | **6.37 (2.2)** | $\propto= $ **0.014** |
| Enjoyment | 4.69 (2.22) | **5.16 (2.4)** | $\propto= $ **0.002** |
| Usefulness | 6.35 (1.9) | **6.4 (2.03)** | $\propto= $ 0.756 |
| Motivation | **7.13 (1.87)** | 6.8 (2.2) | $\propto= $ **0.009** |
| Confidence | **8.4 (1.57)** | 6.9 (1.43) | $\propto= $ 0.826 |
| Procrastination | **6.89 (1.43)** | 6.7 (1.54) | $\propto= $ **0.043** |

The results of Table 1 show favorable results in the dimensions of anxiety and enjoyment, which is to say that at the end of the course the students demonstrated and improved their ability to improve anxiety and indicated an increased liking of mathematics; however, in the dimensions of motivation and procrastination, the results were not favorable, because at the beginning of the semester the students perceived themselves to more motivated by mathematics than at the end; similarly, their resources to avoid procrastination decreased. Table 2 shows the correlations.

**Table 2.** Dimension correlations.

| | Anxiety | Enjoyment | Usefulness | Motivation | Confidence | Procrastination |
|:---:|:---:|:---:|:---:|:---:|:---:|:---:|
| Anxiety | 1 | 0.559 ** | 0.529 ** | 0.401 ** | 0.381 ** | 0.543 ** |
| Enjoyment | | 1 | 0.661 ** | 0.224 ** | 0.505 ** | 0.434 ** |
| Usefulness | | | 1 | 0.408 ** | 0.526 ** | 0.397 ** |
| Motivation | | | | 1 | 0.328 ** | 0.369 ** |
| Confidence | | | | | 1 | 0.332 ** |
| Procrastination | | | | | | 1 |

** $\alpha = 0.01$.

It can be observed that the correlation between the dimensions is positive but not strong; the highest correlation is between usefulness and enjoyment; however, in the usefulness dimension, no significant difference was found. However, when the results were analyzed by category, including professor, scholarship, gender, program (bicultural or

multicultural) and semester, the following findings were obtained. Professor 1 obtained a significant favorable difference in the anxiety dimension ($\propto= 0.21$) and ($\propto= 0.029$) enjoyment; and professor 8 was close in the anxiety dimension ($\propto= 0.051$). The rest of the professors showed no significant difference. With respect to scholarship status, non-scholarship students showed an unfavorable result in the motivation and ($\propto= 0.001$) procrastination dimensions ($\propto= 0.016$), while scholarship students showed favorable results in the anxiety ($\propto= 0.041$) and enjoyment dimension ($\propto= 0.002$).

With respect to gender, both women and men showed significant differences in the satisfaction dimension, although ($\propto= 0.020$) men had an unfavorable result in the motivation dimension ($\propto= 0.025$). In the semester category, it was found that younger students showed an improvement in the anxiety ($\propto= 0.01$) and enjoyment dimensions ($\propto= 0.033$), while the result in the motivation dimension decreased ($\propto= 0.020$) ($\propto= 0.013$). In the rest of the semesters, no significant differences were found in any dimension.

Finally, with respect to the program, the students of the multicultural program did not show significant changes between the pre-test and post-test, while the students of the bicultural program showed significant favorable differences in the anxiety dimension and ($\propto= 0.019$) enjoyment ($\propto= 0.001$), as well as significant unfavorable differences in the motivation ($\propto= 0.011$) and procrastination ($\propto= 0.033$) dimensions. The means are shown in Table 3.

**Table 3.** Means and deviation standards of significative results.

| Category | Pre-Test Mean | Post-Test Mean |
|---|---|---|
| Professor 1 (anxiety) | 5.92 (2.07) | 6.67 (2.08) |
| Professor 1 (enjoyment) | 4.95 (2.14) | 5.74 (2.51) |
| Professor 8 (anxiety) | 6.03 (1.92) | 6.69 (2.05) |
| No scholarship (motivation) | 6.93 (1.84) | 6.28 (2.04) |
| No scholarship (procrastination) | 6.76 (1.33) | 6.42 (1.5) |
| Scholarship (anxiety) | 6.36 (2.12) | 6.77 (2.19) |
| Scholarship (enjoyment) | 4.93 (2.23) | 5.59 (2.38) |
| Scholarship (motivation) | 7.02 (1.96) | 6.5 (1.88) |
| Semester 1(anxiety) | 5.97 (2.17) | 6.39 (2.27) |
| Semester 1(enjoyment) | 4.95 (2.28) | 5.44 (2.44) |
| Semester 1(motivation) | 7.27 (1.87) | 6.82 (2.18) |
| Bicultural program (anxiety) | 5.98 (2.15) | 6.36 (2.24) |
| Bicultural program (enjoyment) | 4.6 (2.21) | 5.15 (2.44) |
| Bicultural program (motivation) | 7.10 (1.91) | 6.72 (2.2) |
| Bicultural program (procrastination) | 6.9 (1.42) | 6.67 (1.52) |

It is interesting to see that in the categories of male and female students, scholarship holders, first income, and the bicultural program obtained better results in the anxiety and pleasure dimensions, and decreased in the motivation dimension in the categories man, non-scholarship, first income, and bicultural program.

Additionally, the hours of study that the students dedicated at the beginning and end of the course were obtained, and the results are shown in Figure 2.

It can be seen that at the end of the course the students dedicated more hours of study than they did at the beginning of the course.

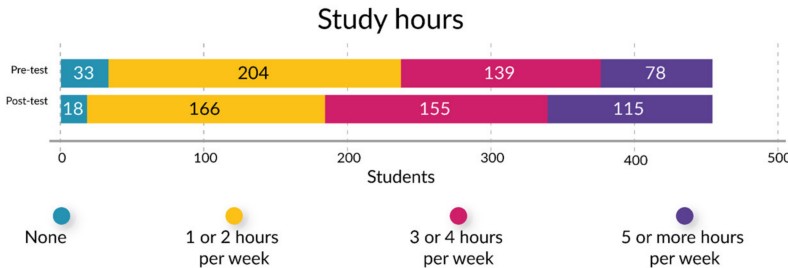

**Figure 2.** Student study hours.

Professor Focus Group

As mentioned, a focus group was conducted to analyze professors' experiences on the five dimensions guided by questions to emphasize and make clear professors' answers.

The first question was about how professors implemented gamification to set a point of connection and differences. Results on this issue showed a minimal but important variability in gamification, as shown in Table 4. All groups were conducted under the superhero's narrative. Groups had a leaderboard for each evaluation period (partial), so the badges count began again after each period during the semester. This practice allowed professors to make adjustments to their gamified parts of the course, and students had the opportunity to participate and make changes to their strategies.

**Table 4.** Variability on gamification implementation.

| Variable | Professor 1 | Professor 2 | Professor 3 | Professor 4 | Professor 5 |
|---|---|---|---|---|---|
| How can students earn a badge? | Spontaneous class participation. Exams' grade progress. | First place in competitions (Khaoot, Socrative, Jeopardy, etc.), reviews, and extra work. | Badges were not necessarily related to the description. Students had to do extra work to earn badges. Also, class participation counted. | Extra weekly activities. Delivering before the due date. Interesting questions from students. Real and considerable progress from one subject to another. | Extra work. Class participation. |
| Most popular or given badge | Flash Stark Grut Fantastic 4 | Not specified | Spiderman or Superman. Flash. Capitan America. | Flash Ironman | Not specified |
| Badges exchange | 5 points on the exam | 10 points for partial (exam) | Maximum 10 points per partial. | Ironman is worth 2 points. | Not specified |
| Leaderboard socialization | By quintiles. Professor showed the leaderboard in class. Students participated, even on WhatsApp. | Professor showed the leaderboard in class when it was updated. | Professor showed the complete upgraded leaderboard. | Not specified | Professor showed the leaderboard in class. |

The professors expressed that students were willing to participate or even do extra activities to have the benefits of their participation on the leaderboard. It is interesting that the most common way to participate was in class but also on already gamified resources such as Kahoot. One of the participant professors combined gamification with Gamit! with the use of class stamps, and students gained badges when they accumulated a set of stamps.

For the professors, there is no doubt about students' motivation in their math class. Table 5 summarizes professors' impressions on this dimension.

**Table 5.** Dimension: attitudes towards mathematics from the focus group.

| Variable | Professors' Perceptions | Comments |
|---|---|---|
| Motivation | "More than obvious". | Participation as evidence. Time looks not to pass on class. |
| Enjoyment | They both (professors and students) enjoy the class more than before. | "I am not sure if it is only gamification, there is a logistics to convince them". |
| Usefulness | Students ask for extra exercises to get badges, but not sure students understand how to apply math in their life. | "They increasingly manipulate their strategies to have more badges". |
| Confidence | Students are more confident, among other topics, because they know they have badges. | "It gave them a lot of confidence to see each other and participate". |
| Anxiety | It has improved in students because they want badges. | "Anxiety has gone up because they are looking for how to climb, but I see it more as giving them that energy to move". |
| Procrastination | For professors who give badges in class, they do not see a relationship, but the professors who give badges for delivery before the due date have noted lowered procrastination. | "Procrastination has not promoted it". "It gives them badges if they deliver before the date". |

### 2.2.2. Second Question Results

We now address the question: What is the relationship between the Gamification methodology based on a reward mechanic and the attention, engagement, and resilience dimensions on the part of students and professors?

First, the descriptive were calculated, as shown in Figure 3.

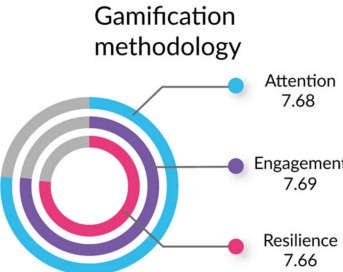

**Figure 3.** Gamification Methodology Dimensions Mean.

We can see that the mean of the three dimensions was similar, and when the ANOVAS were calculated by categories, no significant differences were found by professor, program, or semester; however, there were differences by gender and scholarship. The results are shown in Table 6.

**Table 6.** ANOVAS by gender and scholarship.

| Categories Dimension | Attention | Engagement | Resilience |
|---|---|---|---|
| Woman-Man | 8.2 (2.1),7.3 (2.4), $\propto= 0.001$ | 8.2 (2.02), 7.3 (2.5), $\propto= 0.001$ | 8.15 (2.14),7.3 (2.6), $\propto= 0.001$) |
| No scholarship/scholarship | 7.4 (2.5),7.93 (2.2) $\propto= 0.028$ | | |

Students were also asked what kind of emotion they felt when they won a badge, and the answers shown in Figure 4.

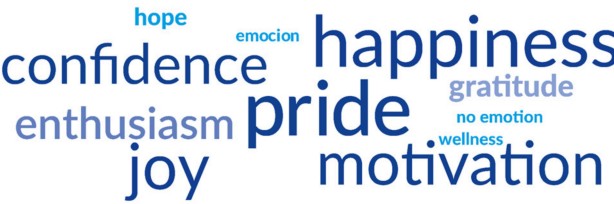

**Figure 4.** Student's emotions.

The emotions of pride, joy, happiness, and motivation were the most voted on, while the words well-being and non-emotion were the least voted on. A recurring comment by students in the open questions was: "The badges allow us to benefit from the points or rewards system while maintaining motivation and healthy competitiveness in the classroom".

### Professor Focus Group

On the dimensions of Attention, Engagement, and Resilience, the professors commented on the following during the focus group session.

1.  Attention

Professors agreed that students were willing to participate if they saw the benefits of using badges There were cases in which students did not have the enthusiasm professors expected to participate on the leaderboard at the beginning of the course. Surprisingly, students were active in class, even proposing activities to get badges. Professors noticed

increased participation from students who did not participate before. Sometimes professors had to guide students or even ask specific students directly to balance the participation and give badges to everyone by merit. Collaboration also improved; students helped their classmates to climb up the leaderboard. A professor said, and the rest agreed, that students who do not need extra points, are the first to participate, but professors contribute to increasing participation from students who usually do not participate in class. Specifically, professors talked about shy students who were encouraged to participate or help their classmates through this gamification system. As another professor said, "they [students] love the exam improvement badge". For the professor, this is evidence of students' motivation and interest in learning.

2. Engagement

In general, the class environment was nicer and sometimes included laughs. Professors noticed students' positive emotions when they showed the leaderboard. Smiles on students' faces were seen with avatars, badges, and participation. Interestingly, students participated, proposed exercises to get badges, and asked their professors for extra work to get badges, but, moreover, students were attentive to their badge's assignments. Students asked their professors for upgraded ranking positions and demanded to see their badges.

On this dimension, professors were not sure whether students were more engaged with the course or the platform, but they did see differences in quality and an opportunity to reinforce other skills and behavior. Professors asked themselves if they can contribute to more platform engagement by frequently showing the leaderboard in class.

3. Resilience

This dimension was divided into three questions to get a clear idea of how professors interpreted their students' resilience. First, professors expressed that they do not see a clear relation between the use of gamification, Gamit! and resilience, but opinions were different about the idea of getting a badge. These dimension answers are more related to engagement because they only talked of expressions and emotions when students got badges and thought that there are other chances to get badges. One professor specified that she saw students' frustration when they did not earn badges, or expressions of "I made it" when they won. This is why she balanced participation in class, asking students who do not participate on their own. Another professor said his group really enjoyed the competition.

On the other hand, a professor expressed that there can be frustration from not earning a badge, but it is only a game. Other professors said students are getting badges not because of mathematical ability but rather as a result of attitude and participation.

The professors agreed that they can help to promote resilience when it is their turn to populate the leaderboard. However, at first sight, they do not see a relation between resilience and gamification at this time of the semester. They see other skills and behaviors, such as negotiation and improvement.

2.2.3. Third Question Results

We then addressed the final question: What is the perception of the usability of the Gamit! platform by students and professors?

We found that the average grade given by the students was 76.5. A significant difference was found with respect to gender (women 7.9 (1.54), men 7.4 (1.8)) and scholarship (non-scholarship 7.45 (1.73) $\propto= 0.028$, scholarship 7.8 (1.63), $\propto= 0.028$). Students were also asked if they would recommend the Gamit! platform and about usability of the platform, the answers of which are presented in Figure 5.

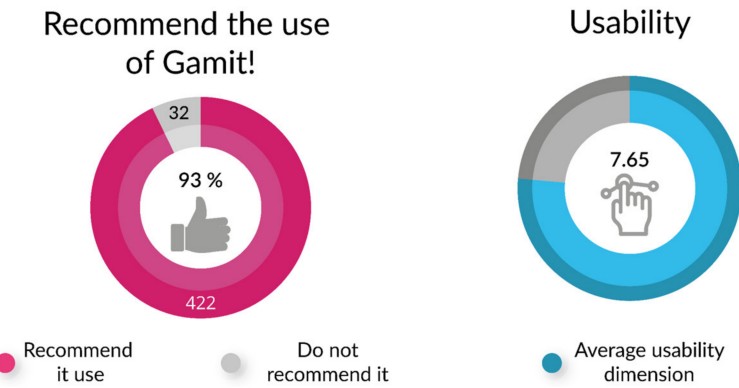

**Figure 5.** Recommendation and usability (students).

Although students would recommend the Gamit! platform, the average rating given for usability still has room for improvement. Among the most positive aspects of the platform were: "Easy to use and check your progress through the ranking", "The use and personalization of avatars motivates us, allows us to identify ourselves quickly and maintain anonymity" and the least positive were: "Improve the design and view of the interface", "that can be accessed from the cell phone".

Professor Focus Group

This dimension was centered on professors' management of gamification and the platform. But it also involves inquiries about students' satisfaction and the perception that it is easy to use.

In general, professors agreed that the platform use helped them to save time on the administration of gamification. They mainly stressed the fact of saving time when assigning badges and showing the leaderboard to the class. Other features were also mentioned, like the automatic ranking and notifications to students who earned badges.

Visual and design aspects were also valued from students' perceptions and professors' experiences. According to professors, students felt attracted to the Gamit! leaderboard compared with the "manual version". Another benefit they see on this gamified experience is the transparency to students, on how and why they got badges and points, making this gamification satisfactory to professors who feel less stress on managing the reward system and making clear for students why they got (or did not get) a badge.

Functionality suggestions regarded the sense of creating a class and the leaderboards, as well as notifications to students.

## 3. Discussion

Regarding attitudes towards mathematics, the general results of the students showed that the dimensions in which there was improvement were in anxiety and enjoyment (see Table 1), which were corroborated by the professors (see Table 5). For [22,23], the improvement of anxiety management promotes a better predisposition in the learning of mathematics, and if we add to this an improvement in the level of enjoyment, undoubtedly the performance of the students could improve. In this sense, gamification based on a reward mechanism can promote an improvement in attitudes towards mathematics in terms of anxiety and enjoyment.

However, the motivation and procrastination dimensions decreased in the case of students (see Table 1), while the results of the focus group showed that professors agreed that they saw students motivated since they actively participated and commented that class time passed very quickly (see Table 5). Regarding procrastination, professors noted that there was not much to do with the reward mechanisms since most of them awarded the badges for participation. In this sense, since there is a discrepancy between students and professors regarding the motivation dimension, it would be necessary to find the factors that provoked the results, particularly since the hours of study reported by students at the

end of the course were greater than at the beginning (see Figure 2). In the study [7,37] it was found that one of the most favored dimensions was motivation, so this result is interesting, and perhaps one of the reasons is that new professors were incorporated in this study.

The relationship of the gamification methodology based on the mechanisms of rewards with respect to the dimensions of attention and engagement were evaluated at almost 77 out of 100 (see Figure 3), and it was found that women and scholarship students rated these dimensions better than men and non-scholarship students (see Table 6). In the focus group, the professors were enthusiastic about gamification and agreed that the students were attentive, participatory, engaged, and manifesting positive emotions. These results coincide with those found in [5,6,11,38,39], in which it is stated that gamification favors attention and engagement. In this sense, gamification based on a reward mechanic and managed by the dynamic platform Gamit! improves attention and engagement in high school mathematics courses.

Finally, regarding the usability of Gamit!, 93% of students said they would recommend it, although with respect to the usability of the platform they gave a rating of 76.5% (see Figure 5), which invites improvement. In addition, the result is encouraging, and several studies indicate that the speed of the learning curve at the beginning of any technology is slower given the resistance to change, but then it improves over time [40,41]. On the other hand, from the professors' perspective, they consider that the platform helps them save time in the management of gamification and that Gamit! allows the allocation of extra points that they made before adopting gamification to be clearer and more fun for students. In this sense, Gamit! [42] is a tool that facilitates the management of gamification for professors, although like all technology it is perfectible and requires time to be adopted.

## 4. Conclusions

From the results of this study, we found that gamification based on a reward mechanism can improve high school students' attitudes towards mathematics, particularly in the dimensions of anxiety and enjoyment. We believe that offering tools within the classroom that help students manage their anxiety about the teaching-learning process can contribute to the benefit of their learning and perhaps in the selection of their professional career. With respect to the motivation dimension that presented a significant unfavorable difference from the students, it invites us to reflect on the relationship between the use of didactic strategies such as gamification and a good teaching-learning process of the math course contents, as well as the meta-evaluation of the strategies applied in the classroom.

Although students perceive the relationship between gamification based on a reward mechanism, and the attention and engagement dimensions as good, professors perceive themselves as being more enthusiastic since they notice that since the gamification of their classes, students have become more participatory, attentive, and even suggest activities to earn badges. They also highlight that the reward mechanism promotes inclusion, since the badges recognize distinct aspects (cognitive, attitudinal, skills and values). Regarding the Gamit! platform, the professors agreed that it facilitates the management of gamification since it saves time and makes it clear to students about the extra points earned.

Finally, the results of the study show that gamification based on a reward mechanic managed by a dynamic platform (Gamit!) favors the learning environment in high school mathematics courses. However, it would be interesting to investigate its effect on learning. On the other hand, there is the possibility of applying the gamification methodology of this study in other disciplines and at other educational levels, always considering that it is a strategy that accompanies a good teaching-learning process and that perhaps gamification is only compatible with those professors who take their teaching practice further.

**Author Contributions:** Conceptualization, E.G.R.-F. and B.N.S.-G.; methodology, E.G.R.-F.; software, E.G.R.-F. and L.M.-C.; validation, E.G.R.-F., B.N.S.-G. and L.M.-C.; formal analysis, E.G.R.-F.; investigation, E.G.R.-F., B.N.S.-G., L.M.-C. and N.K.R.-R.; resources, E.G.R.-F., B.N.S.-G., L.M.-C., N.K.R.-R., H.A.Q.-C. and A.M.-S. data curation, E.G.R.-F., B.N.S.-G. and L.M.-C.; writing—original draft preparation E.G.R.-F., B.N.S.-G., L.M.-C., N.K.R.-R., H.A.Q.-C. and A.M.-S.; writing—review and editing,

E.G.R.-F., B.N.S.-G., L.M.-C. and N.K.R.-R.; visualization, E.G.R.-F., B.N.S.-G. and L.M.-C.; supervision, E.G.R.-F.; project administration, E.G.R.-F., B.N.S.-G., L.M.-C., N.K.R.-R., H.A.Q.-C. and A.M.-S. All authors have read and agreed to the published version of the manuscript.

**Funding:** This research received no external funding.

**Institutional Review Board Statement:** The study was conducted in accordance with the Declaration of Helsinki and approved by the Ethics Committee of the Instituto Tecnológico de Monterrey, México, and Universidad de Lima, Perú.

**Informed Consent Statement:** Informed consent was obtained from all subjects involved in the study.

**Data Availability Statement:** Not applicable.

**Acknowledgments:** The authors are grateful for the support of Writing Lab and Impact Measurement area of the Institute for the Future of Education of Tecnológico de Monterrey, and Universidad de Lima, Peru. Also, to the professors of Prepa Tec Campus Ciudad of Mexico: Jessica I. Vicencio Andrade, Patricia Ramírez Tlalpan, and Brenda E. Delgado Morales, for their collaboration in the research.

**Conflicts of Interest:** The authors declare no conflict of interest.

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
