# Peer review of "Gamit! Icing on the Cake for Mathematics Gamification"

_sustainability, doi:10.3390/su15032334_

Round 1

Reviewer 1 Report

Introduction and Theoretical Framework

This is a growing topic, especially after the pandemic and the need for computers. In addition, the use of gamification allows, as has already been demonstrated, to engage and motivate the student.

Congratulations to the authors for the introduction.

Material and Methods

It is very positive to have the same tool to evaluate and extract the data from the 454 students, allowing it to be unbiased.

In any case, it would be convenient to justify the reason for choosing the questions asked (lines 163-170).

As for the qualitative analysis of the teachers, the questionnaire/test given to them should be systematized.

Results and discussion

The results are well presented, and reflect at all times with great accuracy and precision those intended in the previous section.

Note: The quality of the images/graphs should be improved, and the table captions should provide more information to facilitate interpretation by the reader.

Conclusions 

The conclusions are presented in a correct and orderly manner, always attending to the stated objectives, which also allows extrapolation to other contexts.

Bibliography/References

The references are correct and well used, but it would be necessary to expand them in the discussion, offering other points of view.

Author Response

We appreciate your comments, regards.

Reviewer 2 Report

The research study is very appreciated as the authors have contributed significant knowledge in an innovative strategy of teaching Mathematics “Gamit! The icing on the cake for mathematics gamification”, however, I have the following minor reservations that may enhance the quality of the manuscript if addressed properly:

1.            Objectives and research questions are missing in the abstract. Further, authors should add methodology sections like a detailed account of samples, data collection, and, analysis tools should be elaborated in detail in the abstract section.

2.            Introduction and review of the literature section are very well written. A very good rationale is also presented but it can be strengthened by supporting solid arguments.  

3.            The quality can be enhanced more if the study is linked with supportive theoretical background with authentic argumentations.

4.            Methodology section is very well written.

5.            Data analysis is very good and results are properly tabulated and presented.

To sum up, the study addressed an important topic; and very good contribution to the relevant field.

Author Response

We appreciate your comments, regards.
